# Polyaromatic nanocapsules as photoresponsive hosts in water

Lorenzo Catti[1], Natsuki Kishida[1], Tomokuni Kai[1], Munetaka Akita[1] & Michito Yoshizawa[1]

Molecular containers that provide both stimuli-responsive assembly/disassembly properties and wide-ranging host capabilities in aqueous medium still remain a current synthetic challenge. Herein we report polyaromatic nanocapsules assembled from V-shaped amphiphilic molecules bearing a photoresponsive *ortho*-dianthrylbenzene unit in water. Unlike previously reported supramolecular capsules and cages, the nanocapsules quickly and quantitatively disassemble into monomeric species by a non-invasive light stimulus through structural conversion from the open to the closed form of the amphiphiles. Regeneration of the nanocapsules is demonstrated by light irradiation or heating of the closed amphiphiles. With the aid of the wide-ranging host capability, the photo-induced release of various encapsulated guest molecules (e.g., Nile red, Cu(II)-phthalocyanine, and fullerene $C_{60}$) can be achieved by using the present nanocapsule in water. This feature can furthermore be utilized to switch the fluorescence of encapsulated coumarin guests through their controlled release.

[1] Laboratory for Chemistry and Life Science, Institute of Innovative Research, Tokyo Institute of Technology, 4259 Nagatsuta, Midori-ku, Yokohama 226-8503, Japan. Correspondence and requests for materials should be addressed to M.Y. (email: yoshizawa.m.ac@m.titech.ac.jp)

Water and light are both essential ingredients for life[1,2]. The rational incorporation of such natural resources into synthetic and materials chemistry is an urgent necessity for the development of sustainable modern technologies. Synthetic molecular cages and capsules usable in aqueous media have been intensively studied owing to their intriguing functions such as selective molecular binding, stabilization of reactive species, and modulation of chemical reactions in their cavities[3–8]. In order to extend the usage of water-soluble molecular cages and capsules, the introduction of light-responsive switches, which potentially allow for non-invasive in vivo studies, into the host frameworks is highly desirable[9–11]. There is a large number of reports in the literature on functional host compounds[12,13] possessing bimodal photoresponsive units, such as azobenzene[14–17], dithienylethene[18,19], and anthracene[20,21]. However, the reported photoactive hosts show no or weak molecular binding and releasing abilities and the majority of these hosts are furthermore not suitable for application in water. Herein we present water-soluble nanocapsules (2) composed of V-shaped amphiphilic molecules o-1 bearing a photoreactive polyaromatic unit (Fig. 1a). The photoresponsive nanocapsule encapsulates a wide range of common hydrophobic compounds (e.g., Nile red, Cu(II)-phthalocyanine, and fullerene $C_{60}$) in water and the resultant host–guest composites enable light-triggered guest release in a quantitative fashion, which could be successfully applied as a fluorescence switch in the case of coumarin guests.

Our design for a water-soluble nanocapsule with photoresponsive host functions originates from V-shaped amphiphilic molecules comprising a *meta*-di(9-anthryl)benzene unit with two hydrophilic groups[22,23]. The light-inactive amphiphiles assemble into spherical micellar capsules through π-stacking interactions and the hydrophobic effect[24]. The capsule displays wide-ranging host abilities in water toward various hydrophobic guests, due to the flexible polyaromatic frameworks adaptable to the guest size and shape[25–27]. Thus we envisioned that incorporation of the *ortho*-derivative, capable of undergoing the intramolecular [4+4] photocyclization of the anthracene panels[28–30], into the water-soluble capsule systems could generate a photoresponsive host

with both guest uptake and release functions in water. The key point of present *ortho*-substituted amphiphile o-1 (Fig. 1b) is that the V-shaped polyaromatic moiety provides both photo switching and guest binding abilities, while photoproduct c-1 loses the binding space due to the generated C-C bond linkages (see Supplementary Fig. 1). In addition, closed amphiphile c-1 converts back to o-1 by both photo and thermal stimuli, resulting in re-formation of the original nanocapsule in water.

## Results

**Quantitative formation of a polyaromatic nanocapsule.** Alkanesulfonate-attached, V-shaped amphiphile o-1a was synthesized in four steps starting from 1,2-dimethoxybenzene (see "Methods" section). The bromination of 1,2-dimethoxybenzene and subsequent Negishi cross-coupling with 9-anthrylzinc chloride in the presence of a $PdCl_2(PhCN)_2/P(t\text{-}Bu)_3$ catalyst allowed the formation of a sterically crowded *ortho*-di(9-anthryl)benzene derivative in a satisfactory yield (68%). Sequential demethylation with $BBr_3$ and etherification using 1,3-propanesultone subsequently gave rise to amphiphile o-1a in 68% yield (over 2 steps). Stirring yellow solid o-1a (2.0 μmol) in water (2.0 ml) in the dark led to the quantitative formation of nanocapsule 2a within 5 min at room temperature. The structure of 2a, consisting of small spherical assemblies $(\text{o-1a})_n$ with a narrow size distribution ($n = $ ~4–6), was confirmed by nuclear magnetic resonance (NMR; see Supplementary Figs. 9 and 10), atomic force microscopy (AFM), and dynamic light scattering (DLS) analyses. In the ¹H NMR spectrum, the anthryl signals ($H_{a-e}$) of 2a in $D_2O$ were significantly broadened and shifted upfield relative to those of o-1a in DMSO-$d_6$ (Fig. 2a, b), indicating effective stacks of the V-shaped polyaromatic frameworks via self-assembly. The AFM measurement of 2a (1.0 mM based on o-1a) under dry conditions on a mica surface revealed the presence of only small spherical particles with an average outer diameter of 2.4 nm (Fig. 3a–c). The particle size and its narrow distribution were also confirmed by the DLS analysis of 2a in $H_2O$ ($d = 1.9$ nm; see Supplementary Fig. 11). The data of the structural analysis corresponds well with

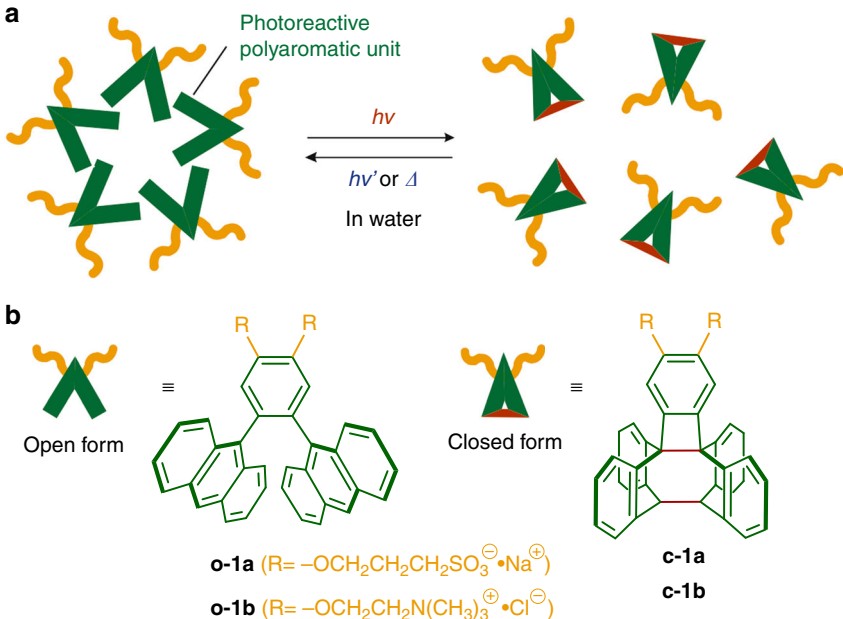

**Fig. 1** Concept and molecular design of a photoresponsive nanocapsule. **a** Schematic representation of a photoresponsive nanocapsule composed of V-shaped amphiphiles bearing a photoreactive *ortho*-dianthrylbenzene unit. **b** Chemical structures of V-shaped amphiphiles o-1a, **b** and their closed forms c-1a, **b** reported herein

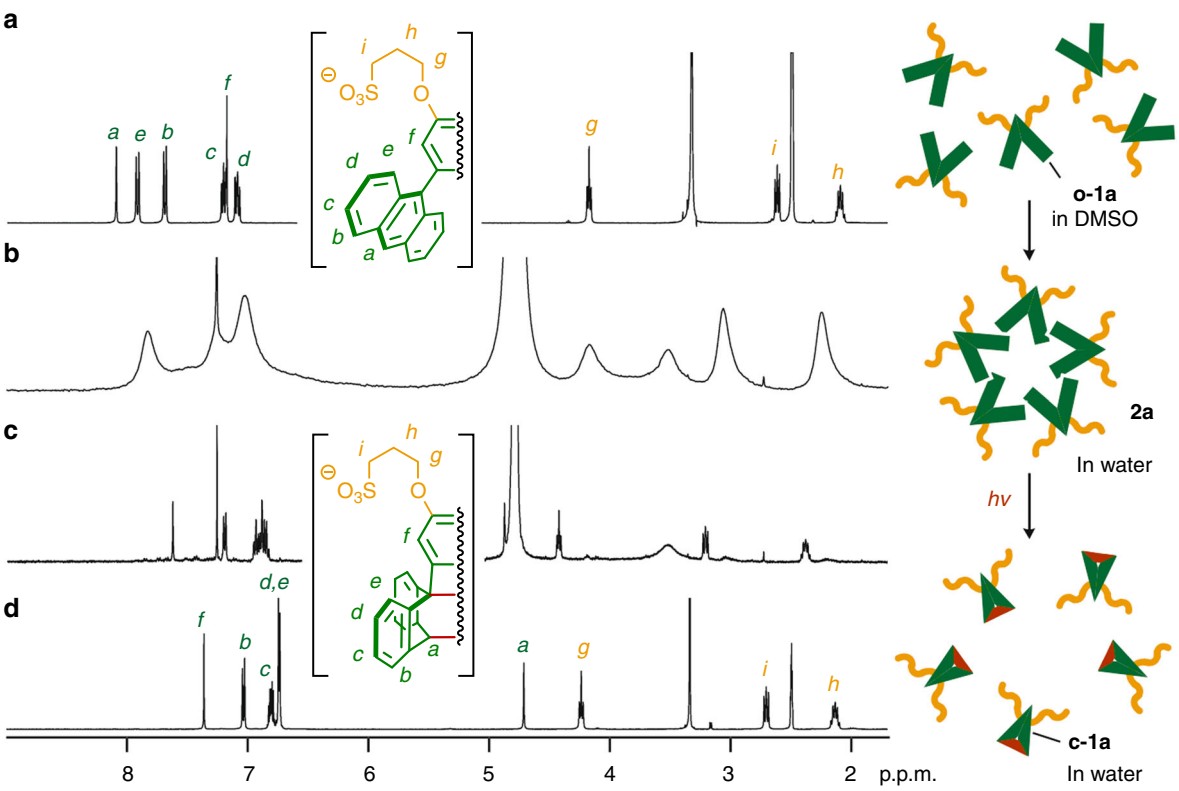

**Fig. 2** Assembly and light-induced disassembly of the nanocapsule in water. ${}^1$H NMR spectra (400 MHz, room temperature; left) and schematic representation (right) of **a** V-shaped amphiphile **o-1a** in DMSO-$d_6$, nanocapsule **2a** (1.0 mM based on **o-1a**) in D$_2$O (with tetramethylsilane in CDCl$_3$ inlet) **b** before and **c** after light irradiation (380 nm) for 10 min and **d** the photoproduct (closed amphiphile **c-1a**) in DMSO-$d_6$

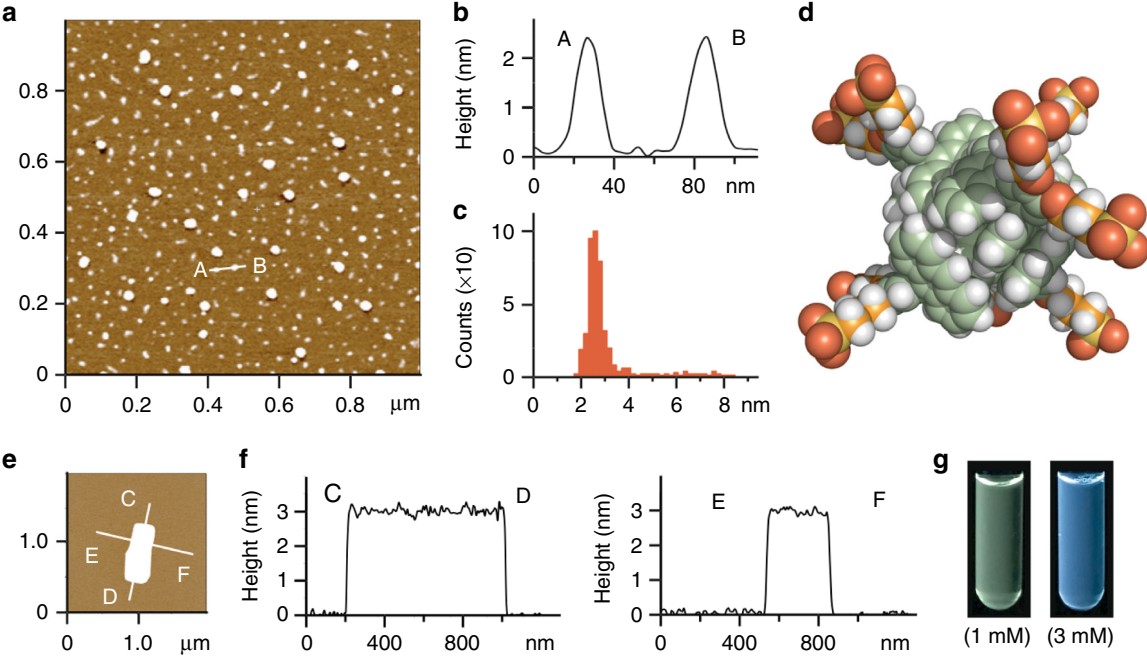

**Fig. 3** Concentration-dependent structural analysis of the nanocapsule. **a** Representative AFM image (room temperature, dry, mica) of nanocapsule **2a** (1.0 mM based on **o-1a**), **b** the selected height profile, and **c** the size and number distribution. **d** Optimized structure of **2a** comprising (**o-1a**)$_5$. **e** Representative AFM image (room temperature, wet, mica) of a rectangular sheet formed by amphiphile **o-1a** (3.0 mM) and **f** the corresponding height profiles. **g** Photographs of the 1.0 mM (left) and 3.0 mM (right) solutions of **o-1a** ($\lambda_{ex} = 365$ nm) in H$_2$O

the structure predicted by molecular modeling for spherical assembly (o-1a)$_5$. The optimized structure of nanocapsule (o-1a)$_5$ provides average core and outer diameters of ~1.8 and ~3.0 nm, respectively (Fig. 3d and see Supplementary Fig. 12). Trimethylammonium-based amphiphile o-1b (Fig. 1b) and its nanocapsule 2b, which is likewise composed of assembled (o-1b)$_n$ (n = ~5), were prepared in a manner similar to o-1a and 2a (see Supplementary Figs. 15–28 and 36). The numeric composition of nanocapsules 2a and 2b is thus comparable to the composition of the previously reported *meta*-derivatives (n = ~5)[22,23].

The fluorescence spectrum of an aqueous solution of 2a (1.0 mM based on o-1a) displayed a broad emission band ($\lambda_{\max} = 535$ nm, $\Phi_F = 3\%$) upon irradiation at 370 nm (see Supplementary Fig. 29b). Interestingly, when the concentration of amphiphile o-1a was increased to 3.0 mM, the emission color of the solution was changed from green to blue (Fig. 3g) and the emission maximum was significantly blue-shifted ($\Delta\lambda = -75$ nm), indicative of a structural transition (see Supplementary Figs. 30 and 31). The observed, prominent Tyndall effect of the 3.0 mM sample suggested the formation of larger assemblies (see Supplementary Fig. 32a). The AFM analysis elucidated the formation of rectangular sheets with lateral sizes of ~350–900 nm and thicknesses of ~3 nm under both wet (aqueous solution) and dry conditions (Fig. 3e, f and see Supplementary Figs. 33 and 34). The colloidal solution was stable enough for at least 1 day at room temperature (see Supplementary Fig. 35). It is noteworthy that the corresponding *meta*-derivative[23] did not display analogous fluorescence behavior in the investigated concentration range (see Supplementary Fig. 31b), highlighting the influence of the substitution pattern on the aggregation behavior of the polyaromatic amphiphile.

The stability of nanocapsule 2a against methanol and temperature was investigated using $^1$H NMR analysis. Formation of nanocapsule 2a is mainly derived from the hydrophobic effect and π-stacking interactions so that the addition of polar organic solvents results in dissociation of the capsular structure. A $^1$H NMR titration study of a D$_2$O solution of 2a (1.0 mM based on o-1a) revealed that around 40% CD$_3$OD by volume are required for complete disassembly into monomers o-1a (see Supplementary Fig. 13). The complete disassembly of 2a was also observed upon heating the solution to around 100 °C (see Supplementary Fig. 14)[31]. In both cases, the gradual dissociation is accompanied with a sharpening of the $^1$H NMR spectrum and a downfield shift of the lateral anthracene protons, caused by the loss of intermolecular π-stacking interactions.

**Reversible assembly–disassembly of the nanocapsule**. Nanocapsule 2 provides highly photoreactive frameworks so that the quick dissociation of 2 into monomeric closed amphiphiles c-1 was successfully achieved in water upon light irradiation. The broadened $^1$H NMR signals of 2a converted to the relatively sharp signals of c-1a in D$_2$O upon 380-nm light irradiation (3 W × 2) for 10 min at room temperature (Fig. 2b, c). The NMR spectral changes as well as the DLS chart of the product indicate the disassembly of the nanocapsule in water (see Supplementary Figs. 41 and 42). Monitoring of the reaction by means of $^1$H NMR analysis furthermore suggested a stepwise disassembly process and the formation of intermediary assemblies (o-1a)$_n$•(c-1a)$_m$ with a reduced degree of π-stacking interactions in water (see Supplementary Fig. 43). The $^1$H NMR spectrum of the photoproduct in DMSO-$d_6$ revealed the quantitative conversion from o-1a to c-1a (Fig. 2d). Similarly, nanocapsule 2b with pendant trimethylammonium groups disassembled quantitatively into monomers c-1b in water under similar conditions (see Supplementary Figs. 46–49).

Furthermore, the reversible assembly–disassembly of the polyaromatic nanocapsule was established by ultraviolet (UV)-visible spectrometry. Nanocapsule 2a showed absorption bands derived from the anthracene panels around 310–440 nm (Fig. 4a), which were slightly red-shifted as compared with those of o-1a in CH$_3$OH (see Supplementary Fig. 29a). When the H$_2$O solution of 2a in a quartz cell was irradiated with UV light (380 nm), the quantitative formation of c-1a was observed after 5 min. The resultant UV-visible spectrum showed no absorption band relative to the anthracene panels (Fig. 4a and see Supplementary Fig. 39), indicating disassembly of the closed amphiphiles in water. It should be noted that nanocapsule 2a was regenerated in 77% yield upon light irradiation of c-1a in H$_2$O at 287 nm for 25 min through structural conversion from c-1a to o-1a (Fig. 4a and see Supplementary Fig. 40). The photo-induced disassembly–assembly cycle was repeated five times under air without a significant degree of decomposition (Fig. 4b). Such a degree of photo reversibility has not been reported so far with non-substituted o-1 (R = -H)[30]. Thermal stimulus converted c-1a into o-1a even more efficiently, thereby regenerating nanocapsule 2a quantitatively from c-1a in H$_2$O at 160 °C for 30 min under microwave conditions (Fig. 4c). The disassembly and assembly processes were also repeated for five times without any sign of decomposition by the light and heat stimuli, respectively (Fig. 4d and see Supplementary Fig. 45).

**Uptake and release of hydrophobic guests using the nanocapsule**. The polyaromatic cavity provided by nanocapsule 2 was subsequently revealed to enable solubilization of a wide variety of sparingly water-soluble, hydrophobic compounds upon encapsulation in water under neutral conditions at room temperature. As a typical example, rapid guest uptake was achieved by manually grinding a 1:1 mixture of solid amphiphile o-1b (0.45 μmol) and Nile red (NR; 0.45 μmol) for 2 min, followed by addition of water (4.5 ml) and removal of excess suspended guest via filtration (Fig. 5a, left). The UV-visible, NMR, and DLS analyses of the resultant, clear red solution indicated the formation of host–guest composite 2b•(NR)$_2$ in a quantitative fashion with respect to o-1b. The UV-visible spectrum showed prominent guest absorption bands in the range of 420–650 nm (Fig. 5b), due to guest uptake into the hydrophobic host cavity. Besides the $^1$H NMR integrals (see "Methods" section), particle size analysis via DLS yielded an average diameter of 2.4 nm for the product, which, in combination with molecular modeling, indicates the formation of a spherical (o-1b)$_6$•(NR)$_2$ structure in average (Fig. 5f and see Supplementary Figs. 51 and 52a). The obtained host–guest structure (even at 0.1 mM based on o-1b) remained intact in water at room temperature for >1 week in the dark.

Importantly, a non-invasive light stimulus allowed the host–guest composite to quantitatively release the encapsulated cargo from the container. Irradiation of the aqueous solution of 2b•(NR)$_2$ for 10 min at 380 nm resulted in complete conversion of amphiphile o-1b into the closed form, c-1b (Fig. 5a, right), as evidenced by UV-visible analysis. Closed amphiphile c-1b loses the V-shaped polyaromatic-binding pocket, which is required for efficient host–guest π/CH-π interactions as well as hydrophobic effects, and therefore is incapable of solubilizing the hydrophobic guests in water. After storage for 1 h at room temperature, the suspended guest aggregates were separated by a sequence of centrifugation (16,000 × $g$, 10 min) and filtration (200 nm in pore size), giving rise to a clear colorless solution containing only c-1b. Quantitative separation of released NR was confirmed by UV-visible analysis (Fig. 5b), showing complete disappearance of the guest absorption bands.

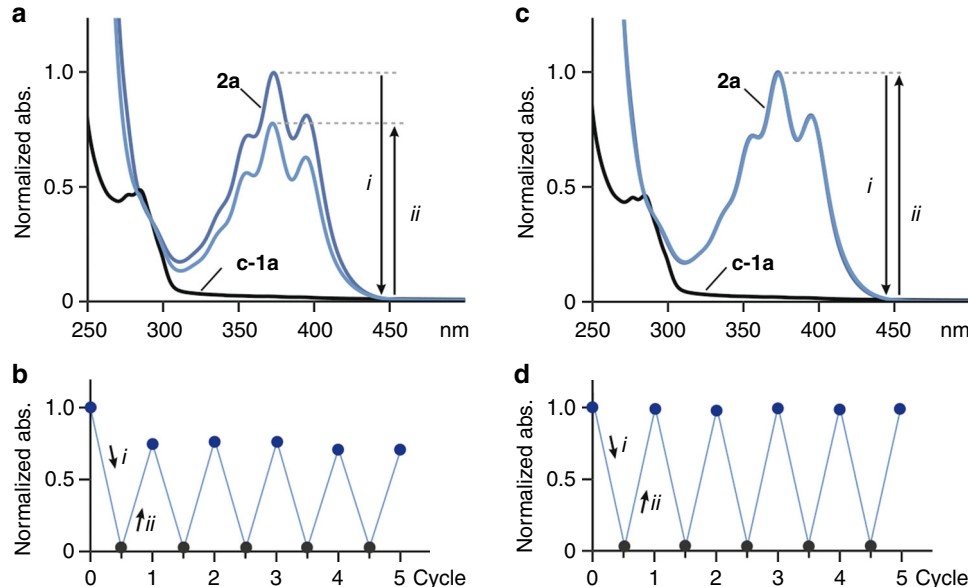

**Fig. 4** UV-visible study of the reversible assembly–disassembly of the nanocapsule. **a** UV-visible spectral changes (H$_2$O, room temperature, 1.0 mM based on **o-1a**) of **2a** upon light irradiation (*i*) at 380 nm for 5 min and then (*ii*) at 287 nm for 25 min (using a fluorescence spectrophotometer (950 V)) and **b** disassembly–assembly cycles of **2a**, monitored by UV-visible spectroscopy (plots of the absorption intensities at 373 nm). **c** UV-visible spectral changes (H$_2$O, room temperature, 1.0 mM based on **o-1a**) of **2a** upon (*i*) light irradiation at 380 nm for 5 min and then (*ii*) heating at 160 °C for 30 min and **d** disassembly–assembly cycles of **2a**, monitored by the UV-visible spectroscopy (373 nm)

The structural flexibility of the nanocapsule likewise enabled the uptake and release of a variety of highly hydrophobic compounds, such as planar Cu(II)-phthalocyanine (**CP**) and Zn (II)-tetraphenylporphyrin (**ZT**), bowl-shaped subphthalocyanine (**SP**), and spherical fullerene C$_{60}$ (**C$_{60}$**), in water. In a manner similar to **2b•(NR)$_2$**, the grinding protocols using a mixture of **o-1b** with **CP** or **C$_{60}$** afforded the corresponding host–guest composites with an approximate host–guest ratio of **2b•(CP)$_m$** (*m* = 2–3) and **2b•C$_{60}$**, respectively. The efficient uptake and quantitative release were verified by UV-visible analysis (Fig. 5c, d). Compared to **NR**, both guests **CP** and **C$_{60}$** required extended aggregation time (2 and 48 h, respectively) for complete separation after release, presumably due to weak interactions between the closed amphiphiles and the extremely hydrophobic guests. Similarly, uptake of water-insoluble **ZT** and **SP** and their quantitative release were accomplished (see Supplementary Fig. 53). Altogether, facile and efficient release of several hydrophobic compounds into bulk water was demonstrated using photoresponsive nanocapsule **2b**. These results furthermore highlight that the host capability of the polyaromatic nanocapsule is not significantly affected by changing the connectivity of the anthracene panels from *meta* to *ortho*[22,23].

**Fluorescence switching of coumarins using the nanocapsule**. Finally, photoresponsive fluorescence switching could be demonstrated using nanocapsule **2a** in water via the uptake and release of emissive coumarin dyes. Grinding a 3:1 mixture of amphiphile **o-1a** and hydrophobic coumarin 314 (**C314**) for 5 min followed by addition of water, 10 min sonication (35 kHz, 100 W), and removal of excess guest gave rise to a clear yellow solution of **2a•(C314)$_2$**. The absorption spectrum displayed a new broad band around 450 nm, assignable to encapsulated (**C314**)$_2$ (see Supplementary Figs. 54 and 56a). Fluorescence quantum yield analysis of the product indicated efficient emission quenching of **C314** within the nanocapsule ($\Phi_F$ = 7%, $\lambda_{ex}$ = 445 nm), due to strong interactions of the guest with the polyaromatic host shell (Fig. 5e). Light irradiation of **2a•(C314)$_2$** at 380 nm for

6.5 min under N$_2$ atmosphere and subsequent filtration of the resultant solution yielded a green fluorescent solution containing **c-1a** and released **C314** as small aggregates, with relatively high quantum yield ($\Phi_F$ = 68%, $\lambda_{ex}$ = 445 nm) and intense emission band at $\lambda_{max}$ = 491 nm (Fig. 5e). Under similar conditions, the emission of coumarin 445 (**C445**) was dramatically enhanced upon irradiation of **2a•(C445)$_2$** in water at 380 nm ($\Delta\Phi_F$ = +71%, $\lambda_{ex}$ = 397 nm, see Supplementary Fig. 55). Notably, host–guest composite **2a•(C314)$_2$** was regenerated in 59% yield (based on the intensity of the guest absorption at 450 nm) via reopening of amphiphile **c-1a** under microwave conditions (30 min at 160 °C), addition of new **C314**, and subsequent sonication for 30 min (35 kHz, 100 W; see Supplementary Fig. 57).

## Discussion
We have developed stimuli-responsive nanocapsules based on an *ortho*-dianthrylbenzene photoswitch. The nanocapsules quantitatively self-assemble in water from V-shaped polyaromatic amphiphiles that feature the photoswitch integrated into the binding motif. Instant light irradiation results in complete and spontaneous disassembly of the capsule through the structural conversion from the open to the closed form of the amphiphiles. This process can be reversed via thermal and photo stimuli. Moreover, the nanocapsule takes up hydrophobic compounds of various size and shape in water and subsequently releases the guests in a quantitative fashion upon light irradiation. The uptake and release characteristics furthermore enable modulation of the emissivity of fluorescent dyes. The present environmentally benign host–guest system represents a promising platform for future investigations into non-invasive light-controlled delivery of biomolecules and synthetic drugs in aqueous medium. It is furthermore anticipated that visible light-controlled guest release from this system will be realized in the future by exploiting recent breakthroughs in sensitized upconversion[32–35].

## Methods
**General**. NMR: Bruker AVANCE-400/HD500 (400/500 MHz) and JEOL ECA400 (400 MHz) for VT NMR, matrix-assisted laser desorption/ionization–time

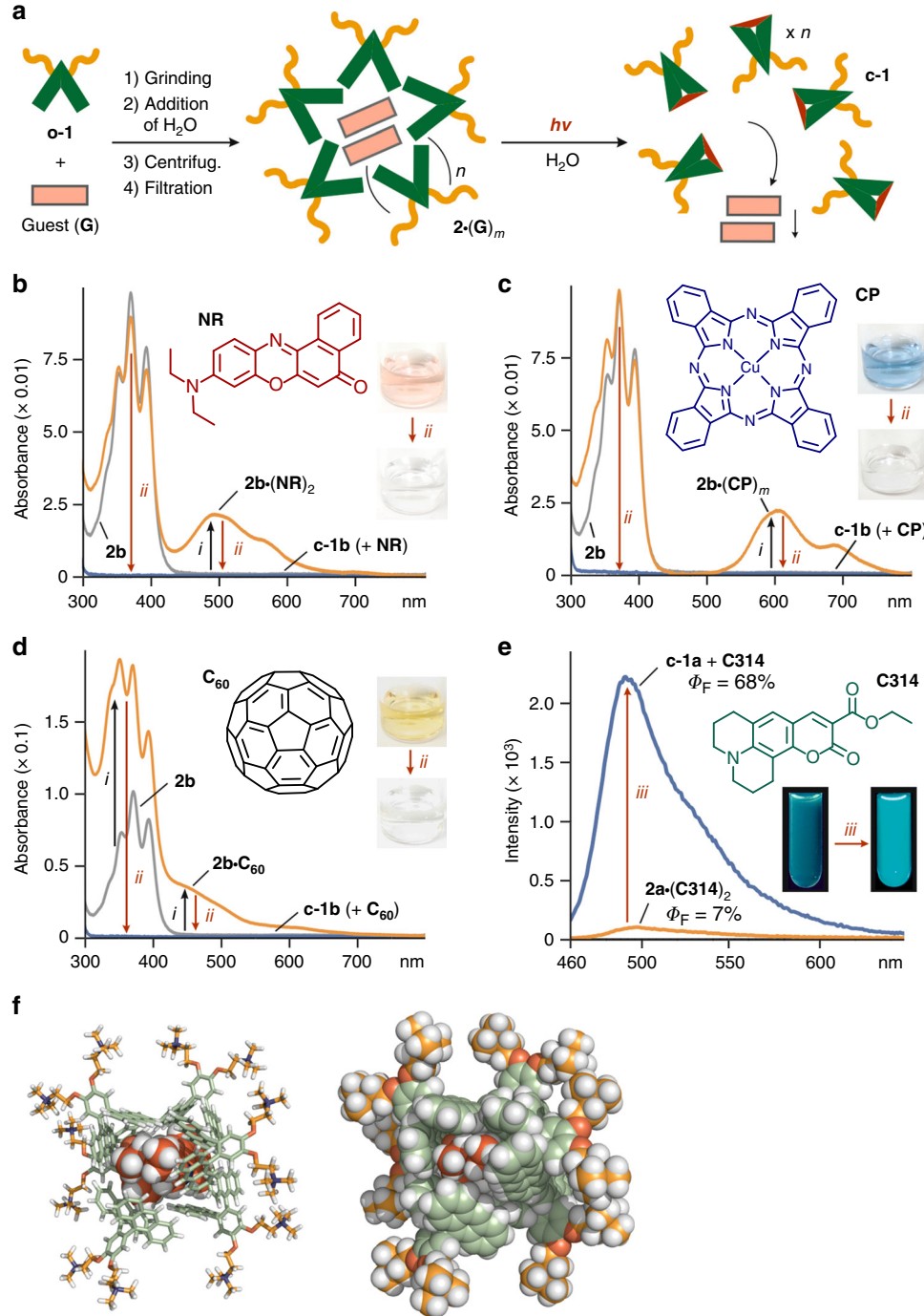

**Fig. 5** Uptake and release of hydrophobic guests by the nanocapsule in water. **a** Schematic representation of the uptake and release of hydrophobic guests (**G**) using photoresponsive nanocapsule **2**. UV-visible spectra (H$_2$O, room temperature, 0.1 mM based on **o-1b**) and photographs of **2b** (*i*) after uptake of **b** NR, **c** CP, and **d** C$_{60}$ and (*ii*) subsequent light irradiation of the corresponding products at 380 nm for 10 min. **e** Fluorescence spectra (H$_2$O, room temperature, 1.0 mM based on **o-1a**, $\lambda_{ex} = 445$ nm), fluorescence quantum yields, and photographs ($\lambda_{ex} = 470$ nm) of **2a•(C314)$_2$** before and after (*iii*) light irradiation at 380 nm for 6.5 min. **f** Optimized structure of **2b•(NR)$_2$**, in which stacked (**NR**)$_2$ are surrounded by six molecules of **o-1b**

of flight (TOF) mass spectrometry (MS): Bruker UltrafleXtreme, electrospray ionization (ESI)-TOF MS: Bruker microTOF II, Fourier transform infrared (FT-IR): SHIMADZU IRSpirit-T, Particle Size Analysis (DLS): Wyatt Technology DynaPro NanoStar, AFM: Asylum Reseach Cypher S, UV-visible: JASCO V-670DS, Emission: Hitachi F7000, Absolute PL quantum yield: Hamamatsu Quantaurus-QY C11347–01, Microwave Reactor: Biotage Initiator+, Lyophilization: Eyela Desktop Freeze Dryer FD-1000, Optimized structure: Accelrys Software Inc., Materials Studio, FORCITE module (version 5.5.3). Solvents and reagents: TCI Co., Ltd., Wako Pure Chemical Industries Ltd., Kanto Chemical Co., Inc., Sigma-Aldrich Co., and Cambridge Isotope Laboratories, Inc. Transfer of liquids with a volume ranging from 10 to 100 µl or from 100 to 1000 µl was performed

with a Nichipet EX Plus II (Nichiryo) pipette equipped with 100 or 1000 µl pipette tips, respectively.

**Synthesis of 1$_{OMe}$.** 9-Bromoanthracene (7.83 g, 30.5 mmol) and dry tetrahydrofuran (THF; 150 ml) were added to a 2-necked 500 ml glass flask filled with N$_2$. A hexane solution (2.6 M) of *n*-butyllithium (11.7 ml, 30.4 mmol) was added dropwise to this flask at –80 °C under N$_2$. After the mixture was stirred at –80 °C for 1 h, a dry THF solution (25 ml) of ZnCl$_2$ (4.50 g, 33.0 mmol) was added to the solution. The resultant mixture was further stirred at –80 °C and then the solution was warmed to room temperature for 1 h to obtain 9-anthrylzinc chloride.

1,2-Dibromo-4,5-dimethoxybenzene (2.99 g, 10.1 mmol), $PdCl_2(PhCN)_2$ (194 mg, 507 μmol), P(t-Bu)$_3$•HBF$_4$ (297 mg, 1.02 mmol), and dry THF (30 ml) were added to a 50 ml glass flask filled with N$_2$[22]. After stirring at room temperature for 1 h, the mixture was added to the 500 ml flask. The resulted solution was further stirred at 80 °C for 15 h. After addition of H$_2$O, the resultant precipitate was collected and washed with CH$_3$OH and hexane to afford 1$_{OMe}$ (3.36 g, 6.86 mmol, 68%) as a white solid (see Supplementary Figs. 2 and 3). The present yield is >2 times higher than that of o-1 (R = -H) by Suzuki-Miyaura cross-coupling[30].

$^{1}$H NMR (400 MHz, CDCl$_3$, room temperature): δ 7.94 (d, $J = 8.7$ Hz, 4 H), 7.89 (s, 2 H), 7.57 (d, $J = 7.8$ Hz, 4 H), 7.28 (s, 2 H), 7.13 (dd, $J = 7.8$, 6.8 Hz, 4 H), 7.03 (dd, $J = 8.7$, 6.8 Hz, 4 H), 4.01 (s, 6 H). $^{13}$C NMR (100 MHz, CDCl$_3$, room temperature): δ 148.3 (C$_q$), 135.4 (C$_q$), 132.1 (C$_q$), 130.7 (C$_q$), 130.0 (C$_q$), 127.9 (CH), 127.5 (CH), 126.2 (CH), 124.4 (CH), 124.2 (C$_q$), 116.2 (CH), 56.2 (CH$_3$). FT-IR (KBr, cm$^{-1}$): 3048, 2948, 2931, 1514, 1462, 1442, 1370, 1246, 1216, 1206, 1093, 1011, 888, 737, 608. HR MS (ESI, CH$_3$OH): $m/z$ Calcd. for C$_{36}$H$_{26}$O$_2$Na [M + Na]$^+$ 513.1825, Found 513.1817.

**Synthesis of o-1a.** Compound 1$_{OMe}$ (1.50 g, 3.06 mmol) and dry CH$_2$Cl$_2$ (40 ml) were added to a 200 ml glass flask. A CH$_2$Cl$_2$ solution (1.0 M) of BBr$_3$ (12 ml, 12.0 mmol) was added dropwise to this flask under N$_2$. The reaction mixture was stirred at room temperature for 7 h. The reaction was quenched with H$_2$O. The product was extracted with EtOAc and the resultant organic layer was dried over MgSO$_4$, filtered, and concentrated. The crude product was dissolved in CHCl$_3$ and then yellow solid 1$_{OH}$ (1.25 g, 2.70 mmol, 88%) was precipitated by the addition of hexane (see Supplementary Fig. 4). Compound 1$_{OH}$ (1.20 g, 2.60 mmol), 60% NaH oil dispersion (339 mg, 8.84 mmol), and dry THF (50 ml) were added to a 200 ml glass flask. 1,3-Propanesultone (940 mg, 7.70 mmol) was added to this flask under N$_2$. The resultant mixture was stirred at 80 °C overnight. The suspension was filtered and the solid residue was washed with hexane. The crude product was dissolved in H$_2$O and then yellow solid o-1a (1.51 g, 2.01 mmol, 77%) was precipitated by the addition of 1-propanol (see Supplementary Figs. 5-8).

1$_{OH}$: $^{1}$H NMR (400 MHz, DMSO-$d_6$, room temperature): δ 9.48 (s, 2 H), 8.08 (s, 2 H), 7.94 (d, $J = 8.8$ Hz, 4 H), 7.69 (d, $J = 7.9$ Hz, 4 H), 7.21 (dd, $J = 7.9$, 7.4 Hz, 4 H), 7.08 (m, 4 H), 7.04 (s, 2 H). ESI-TOF MS (CH$_3$OH): $m/z$ Calcd. for C$_{34}$H$_{21}$O$_2$ [M – H]$^-$ 461.15, Found 461.18.

o-1a: $^{1}$H NMR (400 MHz, DMSO-$d_6$, room temperature): δ 8.09 (s, 2 H), 7.92 (d, $J = 8.8$ Hz, 4 H), 7.69 (d, $J = 8.4$ Hz, 4 H), 7.20 (m, 4 H), 7.18 (s, 2 H), 7.09 (m, 4 H), 4.18 (t, $J = 6.4$ Hz, 4 H), 2.63 (t, $J = 7.6$ Hz, 4 H), 2.11 (m, 4 H). $^{13}$C NMR (100 MHz, DMSO-$d_6$, room temperature): δ 147.8 (C$_q$), 135.2 (C$_q$), 131.0 (C$_q$), 130.2 (C$_q$), 129.4 (C$_q$), 127.6 (CH), 127.3 (CH), 125.7 (CH), 124.6 (CH), 124.4 (CH), 117.9 (CH), 67.9 (CH$_2$), 48.0 (CH$_2$), 25.5 (CH$_2$). FT-IR (KBr, cm$^{-1}$): 3448, 2944, 1648, 1512, 1246, 1201, 1054, 1015, 886, 847, 790, 735, 619, 610, 531. HR MS (ESI, CH$_3$OH): $m/z$ Calcd. for C$_{40}$H$_{32}$O$_8$S$_2$Na [M – Na]$^-$ 727.1442, Found 727.1443.

**Synthesis of c-1a.** A DMSO-$d_6$ solution (2.23 ml) of V-shaped amphiphilic molecule o-1a (1.67 mg, 2.23 μmol) was irradiated with 380-nm light for 10 min. The quantitative formation of c-1a was confirmed by NMR and MS analyses (see Supplementary Figs. 37 and 38). In contrast, a solid sample of o-1a is relatively stable under UV light irradiation and showed no conversion into c-1a even after irradiation with 380-nm light (3 W × 2) for 10 min (see Supplementary Fig. 44).

$^{1}$H NMR (500 MHz, DMSO-$d_6$, room temperature): δ 7.36 (s, 2 H), 7.03 (d, $J = 7.0$ Hz, 4 H), 6.81 (m, 4 H), 6.77–6.70 (m, 8 H), 4.72 (2, 2 H), 4.24 (t, $J = 6.3$ Hz, 4 H), 2.17 (t, $J = 7.5$ Hz, 4 H), 2.14 (m, 4 H). $^{13}$C NMR (125 MHz, DMSO-$d_6$, room temperature): δ 150.5 (C$_q$), 142.4 (C$_q$), 142.3 (C$_q$), 132.8 (C$_q$), 127.2 (CH), 125.5 (CH), 125.2 (CH), 123.9 (CH), 112.0 (CH), 72.1 (C$_q$), 68.2 (CH$_2$), 51.9 (CH), 48.1 (CH$_2$), 25.5 (CH$_2$). FT-IR (KBr, cm$^{-1}$): 3470, 3069, 2935, 1641, 1492, 1470, 1334, 1196, 1134, 1050, 849, 787, 718, 604. HR MS (ESI, CH$_3$OH): $m/z$ Calcd. for C$_{40}$H$_{32}$O$_8$S$_2$Na [M – Na]$^-$ 727.1442, Found 727.1442.

**Synthesis of o-1b and c-1b.** Compound 1$_{OH}$ (0.959 g, 2.07 mmol), NaOH (4.24 g, 106 mmol), and dry toluene (100 ml) were added to a 2-necked 300 ml glass flask filled with N$_2$. The resultant mixture was stirred at 80 °C for 1 h. 2-Chloro-N,N-dimethylethanamine hydrochloride (3.11 g, 21.6 mmol) was added to the glass flask at room temperature. The resultant mixture was stirred at 125 °C for 1 d[22]. The reaction was quenched with H$_2$O (100 ml). The crude product was extracted with EtOAc (100 ml × 3). The combined organic extracts were dried over Na$_2$SO$_4$, filtered, and concentrated under reduced pressure to afford 1$_{NMe2}$ (1.18 g, 1.95 mmol, 94%) as a yellow solid. Compound 1$_{NMe2}$ (1.18 g, 1.95 mmol) and dry CH$_3$CN (100 ml) were added to a 300 ml glass flask. CH$_3$I (0.75 ml, 12.0 mmol) was added dropwise to this flask. The resultant mixture was stirred at room temperature for 1 day. The solvent was removed under vacuum and then the obtained solid was washed with acetone to give 1$_{NMe3}$ (1.34 g, 1.51 mmol, 77%) as a yellow solid. Compound 1$_{NMe3}$ (1.34 g, 1.51 mmol), AgCl (0.658 g, 4.59 mmol), and H$_2$O (10 ml) were added to a 100 ml glass flask in the dark. The resultant mixture was stirred at 80 °C for 1 day. Then CH$_3$OH was added to the flask and the resultant solution was filtered through a membrane filter to remove AgI. The solvent was removed under vacuum and then the obtained solid was washed with acetone to afford o-1b (0.617 g, 874 μmol, 58%) as a yellow solid. A D$_2$O solution of 2b (0.1 mM) was irradiated with a LED lamp (λ = 380 nm, 3 W × 2) for 10 min to afford c-1b quantitatively.

1$_{OH}$: $^{1}$H NMR (400 MHz, CDCl$_3$, room temperature): δ 7.93 (s, 2 H), 7.91 (d, $J = 8.8$ Hz, 4 H), 7.60 (d, $J = 7.9$ Hz, 4 H), 7.28 (s, 2 H), 7.14 (dd, $J = 7.9$, 7.5 Hz, 4 H), 7.01 (dd, $J = 8.8$, 7.5 Hz, 4 H), 4.23 (t, $J = 5.7$ Hz, 4 H), 2.87 (t, $J = 5.7$ Hz, 4 H), 2.39 (s, 12 H). ESI-TOF MS (CH$_3$OH): $m/z$ Calcd. for C$_{42}$H$_{41}$N$_2$O$_2$ [M + H]$^+$ 605.3, Found 605.2.

1$_{NMe3}$: $^{1}$H NMR (400 MHz, DMSO-$d_6$, room temperature): δ 8.15 (s, 2 H), 7.92 (d, $J = 8.2$ Hz, 4 H), 7.73 (d, $J = 7.9$ Hz, 4 H), 7.45 (s, 2 H), 7.24 (dd, $J = 7.9$, 7.5 Hz, 4 H), 7.11 (dd, $J = 8.2$, 7.5 Hz, 4 H), 4.60 (br, 4 H), 3.83 (t, $J = 4.8$ Hz, 4 H), 3.25 (s, 18 H).

o-1b: $^{1}$H NMR (400 MHz, CD$_3$OD, room temperature): δ 8.00 (s, 2 H), 7.89 (d, $J = 9.0$ Hz, 4 H), 7.64 (d, $J = 7.8$ Hz, 4 H), 7.53 (s, 2 H), 7.15 (dd, $J = 7.8$, 6.8 Hz, 4 H), 7.03 (dd, $J = 9.0$, 6.8, 1.2 Hz, 4 H), 4.69 (br, 4 H), 3.96 (t, $J = 4.6$ Hz, 4 H), 3.37 (s, 18 H). $^{13}$C NMR (100 MHz, CD$_3$OD, room temperature): δ 148.2 (C$_q$), 135.8 (C$_q$), 135.4 (C$_q$), 132.2 (C$_q$), 131.2 (C$_q$), 129.1 (CH), 128.4 (CH), 127.5 (CH), 125.6 (CH), 125.5 (CH), 120.1 (CH), 66.7 (CH$_2$), 64.6 (CH$_2$), 55.0 (CH$_3$). FT-IR (KBr, cm$^{-1}$): 3453, 3048, 2925, 1624, 1571, 1481, 1367, 1248, 1199, 1169, 1100, 960, 736. HR MS (ESI, CH$_3$OH): $m/z$ Calcd. for C$_{44}$H$_{46}$N$_2$O$_2$Cl [M – Cl]$^+$ 669.3242, Found 669.3224.

c-1b: $^{1}$H NMR (400 MHz, CD$_3$OD, room temperature): δ 7.80 (s, 2 H), 7.35 (d, $J = 7.2$ Hz, 4 H), 7.09 (m, 4 H), 7.00–6.79 (m, 8 H), 5.04 (s, 2 H), 4.10 (br, 4 H), 3.50 (s, 18 H). $^{13}$C NMR (125 MHz, CD$_3$OD, room temperature): δ 149.1 (C$_q$), 142.5 (C$_q$), 142.1 (C$_q$), 133.8 (C$_q$), 127.4 (CH), 125.7 (CH), 125.1 (CH), 123.9 (CH), 112.2 (CH), 72.2 (C$_q$), 64.4 (CH$_2$), 63.0 (CH$_2$), 53.4 (CH$_3$), 51.8 (CH). HR MS (ESI, CH$_3$OH): $m/z$ Calcd. for C$_{44}$H$_{46}$N$_2$O$_2$Cl [M – Cl]$^+$ 669.3242, Found 669.3230.

**Formation of nanocapsules 2a and 2b.** Compound o-1a (1.07 mg, 1.43 μmol) and D$_2$O (1.43 ml) were added to a glass test tube. When the mixture was stirred for 5 min in the dark at room temperature, the formation of 2a was confirmed by NMR, UV-visible, fluorescence, DLS, and AFM analyses. Similarly, when a mixture of o-1b (0.28 mg, 0.40 μmol) and D$_2$O (4.0 ml) was stirred at room temperature for 1 min, the formation of 2b was confirmed by NMR, UV-visible, fluorescence, DLS, and AFM analyses.

2a: $^{1}$H NMR (500 MHz, D$_2$O, 1.0 mM based on o-1a, room temperature, tetramethylsilane (TMS) as an external standard): δ 8.51–5.73 (m, 20 H), 4.46–3.47 (br, 4 H), 3.35–2.59 (br, 4 H), 2.48–1.87 (br, 4 H). DOSY NMR (500 MHz, D$_2$O, 1.0 mM based on o-1a, 25 °C): $D = 2.15 \times 10^{-10}$ m$^2$ s$^{-1}$.

2b: $^{1}$H NMR (400 MHz, D$_2$O, 0.1 mM based on o-1b, room temperature, TMS as an external standard): δ 8.11 (d, $J = 7.7$ Hz, 4 H), 7.72 (s, 2 H), 7.61 (s, 2 H), 7.46 (d, $J = 7.6$ Hz, 4 H), 7.32 (dd, $J = 7.7$, 7.1 Hz, 4 H), 7.27 (dd, $J = 7.6$, 7.1 Hz, 4 H), 4.05 (br, 4 H), 3.48 (s, 18 H). DOSY NMR (500 MHz, D$_2$O, 0.1 mM based on o-1b, 25 °C): $D = 3.09 \times 10^{-10}$ m$^2$ s$^{-1}$.

**Encapsulation and release of NR by nanocapsule 2b.** A mixture of amphiphilic compound o-1b (0.32 mg, 0.45 μmol) and NR (0.14 mg, 0.45 μmol) was ground for 2 min using an agate mortar and pestle. After the addition of H$_2$O (4.5 ml), the suspended solution was centrifuged (16,000 × g, 10 min) and then filtered by a membrane filter (pore size: 200 nm) to give a clear pale red solution of 2b•(NR)$_m$. The quantitative formation of 2b•(NR)$_m$ was confirmed by DLS and UV-visible analyses. The average host–guest ratio (o-1b:NR = 3:1) was estimated by integration of the sharp $^{1}$H NMR resonances of the lyophilized product mixture in CD$_3$OD (see Supplementary Fig. 50). In contrast, the $^{1}$H NMR spectrum of 2b•(NR)$_2$ in D$_2$O only shows signals derived from the host framework, which can be explained by the significant broadening of the guest signals due to the restricted motion of the guests within the limited cavity space[22]. The resultant H$_2$O solution was irradiated with a LED lamp (λ = 380 nm, 3 W × 2) for 10 min. The obtained suspension was centrifuged (16,000 × g, 10 min) after 1 h and then filtered by a membrane filter (pore size: 200 nm) to give a clear solution of c-1b. The quantitative release of NR was confirmed by UV-visible analysis. The encapsulation and release of CP, C$_{60}$, SP, or ZT by 2b was performed in a similar way. Sulfonate-based nanocapsule 2a displays comparable host capabilities but inferior release of highly hydrophobic guests, as compared with 2b, likely due to residual interactions between the released guest and closed amphiphile c-1a in water.

## Data availability

The authors declare that the data supporting the findings of this study are available within the Supplementary Information files and from the corresponding author upon reasonable request.

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

## Acknowledgements

This work was supported by JSPS KAKENHI (Grant No. JP17H05359/JP18H01990/JP19H04566) and "Support for Tokyotech Advanced Researchers (STAR)". Professor Kimihisa Yamamoto and Dr. Makoto Tanabe (Tokyo Institute of Technology) are greatly acknowledged for the provision of the microwave reactor. We thank Dr. Motoya Suzuki and Yoshiyuki Satoh (Tokyo Institute of Technology) for help with AFM measurements and experimental support, respectively. L.C. thanks the JSPS and Humboldt Postdoctoral Fellowship.

## Author contributions

L.C., N.K., and M.Y. designed the work, carried out research, analyzed data, and wrote the paper. T.K. carried out research. M.A. was involved in the work discussion. M.Y. is the principal investigator. All authors discussed the results and commented on the manuscript.

## Additional information

**Competing interests:** The authors declare no competing interests.

