## [Peer review file · Nature Communications]

Reviewers' comments:

Reviewer #1 (Remarks to the Author):

This manuscript from the Yoshizawa group describes the generation of nanocapsules in water from amphiphilic aromatic building blocks consisting of a benzene core with two adjacent 9-anthryl substituents. These amphiphiles are structurally related to the meta-disubstituted analogs developed in the same group. While the self-assembly properties of both types of amphiphiles are closely related, only the newly developed ortho-disubstituted derivatives can be reversibly switched by light irradiation (or light irradiation followed by thermal treatment) between an open and a closed structure. Since just the open structure is capable of self-assembly, the assembly/disassembly of the nanocapsules in water can be controlled by an external stimulus. The photo-induced disassembly of the nanocapsules moreover offers a means to release in a controlled fashion water-insoluble organic compounds encapsulated within the nanocapsules.

The concept underlying the formation of analogous nanocapsules in water from meta-disubstituted analogs has been reported before, but the work described in this manuscript develops this idea in a new and very interesting direction. The authors present substantial experimental evidence for the self-assembly of the open form of the ortho-disubstituted amphiphile, they demonstrate the reversible photo-induced assembly/disassembly of the nanocapsules, and they also show that hydrophobic guests encapsulated in the nanocapsules can be released once the building blocks are switched from the open to the closed form. Overall, the work is highly original and the results convincing, which is why publication of this manuscript in Nat. Commun. is strongly recommended. A few minor revisions are, however, necessary.

- The authors should add a few sentences about the synthetic strategy used to prepare the ortho-disubstituted amphiphiles described. The first sentences in the Results and Discussion part are not very clear and insufficient for a reader to understand easily how the compounds were prepared.
- The authors should also refer more precisely to the exact composition of the nanocapsules. The AFM image, the DLS measurements, and also the other methods indicate that the nanocapsules are structurally not uniform but feature a certain size distribution. Thus, the pentamer shown in Figure 3D seems to represent only one of the possible aggregates in solution. The lower and upper number of amphiphiles making up the nanocapsules in dilute solution should be specified.
- Related to the last point is the composition of the nanocapsules with included guests. The authors assign a composition of 2b.(NR)₂ for the Nile red complex, for example, but it is unclear how they arrived at this composition. First, how many amphiphiles does 2b contain (exactly five or five on average or a broader range of values)? Second, is the specified composition an average or an exact composition? How was the composition determined? NMR seems to be less suitable because of the broadening of the signals in the spectrum. According to the Supporting Information, UV-Vis spectroscopy was therefore mostly used but for this, the exact molar attenuation coefficients of the amphiphile and also the guests must be known. Is it possible to estimate the molar attenuation coefficients of the guests when bound (in close proximity) inside the nanocapsules from known values in solution? If approximations were used, how reliable is the composition specified?
- This reviewer was also wondering whether guest encapsulation affects the size/composition of the nanocapsules because the self-assembly is likely dynamic in solution and the number of amphiphiles making up a nanocapsule can therefore adapt to the size or shape of the guest bound.
- Is photo-switching of the nanocapsules affected if a guest is included into the capsules whose UV-Vis band features bands overlapping with the ones of the anthryl residues that need to be excited for the [4+4] cycloaddition?

Reviewer #2 (Remarks to the Author):

The present manuscript reports a new class of photoresponsive nanocapsule whose formation is driven by the aqueous environment. The introduction of a photoresponsive o-dianthrylbenzene in the amphiphilic components leads to photoresponsive assembly-disassembly of the corresponding

nanocapsule. This is the major claim of the paper and the important reason for it to be published in Nature Communications. The manuscript is clearly presented, well-written and contains all the relevant information in terms of characterization, inclusion properties and photochemically driven release of encapsulated guests. The only missing item is the introduction of even larger guests to test the limit of these amphiphiles assembly in terms of internal volume and assembly stoichiometry. The combination of photochemical assembly-disassembly and water solubility makes these nanocapsule appealing for researchers active in the field of drug delivery.

Reviewer #3 (Remarks to the Author):

In this manuscript, Yoshizawa and coworkers presented a very interesting molecular container with light stimuli-responsive behavior. They successfully prepared a novel polyaromatic nanocapsules from V-shaped amphiphilic building blocks bearing a photoresponsive ortho-dianthrylbenzene moiety in water. They found that the resultant capsules can quickly and quantitatively disassemble into monomeric species by a non-invasive light stimulus through structural conversion from the open to the closed form of the amphiphiles. With such supramolecular transformation, the photo-induced release of various encapsulated guest molecules (e.g., Nile red, Cu(II)-phthalocyanine, and fullerene C₆₀) was successfully achieved by using the present nanocapsule in water.

It should be noted that the construction of stimuli-responsive supramolecular functional materials has always been one of the most attractive topics within supramolecular chemistry and materials science. However, the construction of supramolecular capsules that provide both stimuli-responsive assembly/disassembly properties and wide-ranging host capabilities in water still remain a challenge. There is no doubt that the authors employed a very "smart" building block to realize such interesting stimuli-responsive capsules, which will receive much attention from the broad readership of both chemistry and materials community. The manuscript is well-organized and the conclusion is very solid. So I strongly recommend it to be published in Nature Communications after minor revisions.

1. The authors claimed that they obtained the structure 2a in water, which contained spherical assemblies (o-1a)_n (n = ~5). I am wondering if such pentamer is the main product in water. Is it possible to obtain other capsules such as hexamer or the higher-order structures?
2. How about the stability of the resultant capsule 2a? For example, will it disassembly if the temperature or the polarity of solvent is changed?
3. If the pentamer is the main product, what happens during the disassembly process induced by light? Is it possible that five ortho-dianthrylbenzene units undergo the light-induced structural transformation simultaneously or in a step-by-step procedure? It will be very helpful if the authors can provide more information about such supramolecular transformation.
4. Two recent reviews are suggested to be cited in the revised version. One is about the topic of supramolecular transformation (Chem. Soc. Rev, 2016, 45, 2656). Another one is about the stimuli-responsive functional assemblies (Acc. Chem. Res., 2018, 51, 2699). It will be very helpful for readers to learn the related field.

For the comments of Reviewer #1:

... The authors present substantial experimental evidence for the self-assembly of the open form of the ortho-disubstituted amphiphile, they demonstrate the reversible photo-induced assembly/disassembly of the nanocapsules, and they also show that hydrophobic guests encapsulated in the nanocapsules can be released once the building blocks are switched from the open to the closed form. Overall, the work is highly original and the results convincing, which is why publication of this manuscript in *Nat. Commun.* is strongly recommended.

We appreciate having a very positive evaluation of our work from Reviewer #1.

1) The authors should add a few sentences about the synthetic strategy used to prepare the ortho-disubstituted amphiphiles described. The first sentences in the Results and Discussion part are not very clear and insufficient for a reader to understand easily how the compounds were prepared.

According to the reviewer's comment, the details of the synthesis of the ortho-substituted V-shaped amphiphiles were added to the revised text (page 3) as follows: Alkanesulfonate-attached, V-shaped amphiphile **o-1a** was synthesized in four steps starting from 1,2-dimethoxybenzene (see the Methods section). The bromination of 1,2-dimethoxybenzene and subsequent Negishi cross-coupling with 9-anthrylzinc chloride in the presence of a $\text{PdCl}_2(\text{PhCN})_2/\text{P}(t\text{-Bu})_3$ catalyst allowed the formation of a sterically crowded ortho-di(9-anthryl)benzene derivative in satisfactory yield (68%). Sequential demethylation with BBr_3 and etherification using 1,3-propanesultone subsequently gave rise to amphiphile **o-1a** in 68% yield (over 2 steps). In addition, the exact experimental procedure and the analytical data of the ortho-substituted amphiphiles **o-1a** and **o-1b** were added to the Methods section (pages 9-12).

2) The authors should also refer more precisely to the exact composition of the nanocapsules. The AFM image, the DLS measurements, and also the other methods indicate that that the nanocapsules are structurally not uniform but feature a certain size distribution. Thus, the pentamer shown in Figure 3D seems to represent only one of the possible aggregates in solution. The lower and upper number of amphiphiles making up the nanocapsules in dilute solution should be specified.

As correctly mentioned by the reviewer, nanocapsule **2a** is a structurally non-uniform product with a narrow size distribution in water. On the basis of the DLS and AFM analyses, combined with molecular modelling, the pentamer is one of the most likely assemblies. For the structural comparison, we added the optimized structures of the tetramer, pentamer, and hexamer as well as their average diameters to Supplementary Fig. 12. Additionally, the following sentences were added to the main text or modified (page 3,4):

The structure of **2a**, consisting of small spherical assemblies $(\mathbf{o-1a})_n$ with a narrow size-distribution ($n = 4-6$), was confirmed by NMR, AFM, and DLS analyses.

The data of the structural analysis corresponds well with the structure predicted by molecular modeling for spherical assembly $(\mathbf{o-1a})_5$.

The optimized structure of nanocapsule $(\mathbf{o-1a})_5$ provides average core and outer diameters of ~ 1.8 and ~ 3.0 nm, respectively (Fig. 3d and see Supplementary Fig. 12).

Trimethylammonium-based amphiphile **o-1b** (Fig. 1b) and its nanocapsule **2b**, which is likewise composed of assemblies (**o-1b**)_n (n = ~5)...

*The numeric composition of nanocapsules **2a** and **2b** is thus comparable to the composition of the previously reported meta-derivatives (n = ~5)^{22,23}.*

3) Related to the last point is the composition of the nanocapsules with included guests. The authors assign a composition of 2b.(NR)₂ for the Nile red complex, for example, but it is unclear how they arrived at this composition. First, how many amphiphiles does 2b contain (exactly five or five on average or a broader range of values)? Second, is the specified composition an average or an exact composition? How was the composition determined? NMR seems to be less suitable because of the broadening of the signals in the spectrum. According to the Supporting Information, UV-Vis spectroscopy was therefore mostly used but for this, the exact molar attenuation coefficients of the amphiphile and also the guests must be known. Is it possible to estimate the molar attenuation coefficients of the guests when bound (in close proximity) inside the nanocapsules from known values in solution? If approximations were used, how reliable is the composition specified?

Unfortunately, the determination of the exact host-guest composition is quite difficult, like in the case of conventional micelles. Since the obtained host-guest structures are not stable under the conditions applied for mass spectrometry (i.e., ESI/MALDI-TOF MS), we determined the structures by UV-visible, NMR, and DLS analyses, combined with molecular modelling. In the case of **2b**•(NR)₂, the UV-visible spectrum suggested a 3:1 ratio of **o-1b**:NR, but we additionally determined the average ratio exactly using ¹H NMR analysis. The method was added to the Methods section (page 13) as follows: “*The average host-guest ratio (o-1b:NR = 3:1) was estimated by integration of the sharp ¹H NMR resonances of the lyophilized product mixture in CD₃OD (see Supplementary Fig. 50)*”.

In addition, we have described that, on the basis of the estimated host-guest ratio, “*particle size analysis via DLS yielded an average diameter of 2.4 nm for the product, which, in combination with molecular modelling, indicates the formation of a spherical (o-1b)₆•(NR)₂ structure in average (see Supplementary Figs. 51 and 52a)*.” in the main text (page 6).

In a manner similar to NR, we decided the average host-guest ratios for the products including **C314** and **C445** by ¹H NMR analysis. On the other hand, the average host-guest ratios for other products including **CP**, **C₆₀**, **SP**, and **ZT** were estimated by UV-visible analysis (with a calibration curve method) in organic solvents (e.g., toluene and CHCl₃) after the lyophilization of the products, because of the insolubility of the host (i.e., amphiphile **o-1b**) in these solvents. These methods were added to the revised SI (page 35-37).

4) This reviewer was also wondering whether guest encapsulation affects the size/composition of the nanocapsules because the self-assembly is likely dynamic in solution and the number of amphiphiles making up a nanocapsule can therefore adapt to the size or shape of the guest bound.

Yes, the present nanocapsule can encapsulate guests of various size and shape *due to the flexible frameworks*. The size and numeric composition of the nanocapsule are affected by

the encapsulated guests (e.g., see Supplementary Fig. 52 (DLS analyses)). We have already reported related results for a *photo-inactive* capsule composed of *meta*-substituted V-shaped amphiphiles (ref. 22-27). To clarify this point, we modified the related sentence in the introduction part (page 2) as follows: "... The light-inactive amphiphiles assemble into spherical micellar capsules through π -stacking interactions and the hydrophobic effect²⁴. *The capsule displays wide-ranging host abilities in water toward various hydrophobic guests, due to the flexible polyaromatic frameworks adaptable to the guest size and shape*²⁵⁻²⁷."

5) *Is photo-switching of the nanocapsules affected if a guest is included into the capsules whose UV-Vis band features bands overlapping with the ones of the anthryl residues that need to be excited for the [4+4] cycloaddition?*

In all of the guest release studies, the quantitative closing of the V-shaped amphiphile occurred within 10 min at 380 nm-light irradiation, even in the case where the UV-visible absorption bands of the guest overlapped with the host absorption band at 380 nm (e.g., C₆₀ and C445). No definite influence could be observed in this photo-switching study. In addition, the following sentence was added to the revised SI (page 37): "*The disassembly of (o-1a)_n•(C445)_m was, like in the case of C₆₀, not significantly influenced by the guest absorption at 380 nm, which can be explained by the almost complete coverage of the guest by the anthracene panels of 2a.*"

For the comments of Reviewer #2:

The present manuscript reports a new class of photoresponsive nanocapsule whose formation is driven by the aqueous environment. The introduction of a photoresponsive o-dianthrylbenzene in the amphiphilic components leads to photoresponsive assembly-disassembly of the corresponding nanocapsule. This is the major claim of the paper and the important reason for it to be published in Nature Communications. The manuscript is clearly presented, well-written and contains all the relevant information in terms of characterization, inclusion properties and photochemically driven release of encapsulated guests.

We greatly appreciate the very positive evaluation of this work from Reviewer #2.

The only missing item is the introduction of even larger guests to test the limit of these amphiphiles assembly in terms of internal volume and assembly stoichiometry.

The present nanocapsule has been shown to encapsulate guests of various size and shape due to the flexible polyaromatic frameworks. The “internal volume” is created by the encapsulated guest and the “assembly stoichiometry” is also determined by the guest identity, in a manner similar to our previous photo-inactive capsule composed of *meta*-substituted V-shaped amphiphiles (ref. 22). The introduction of even larger guests (e.g., dicoronylene, dumbbell-shaped fullerene, and multi-walled carbon nanotube) has been accomplished by the previous capsule (ref. 24 and 27). Since the host capability of the *ortho*- and *meta*-substituted capsules are quite similar, the encapsulation of even larger guests should be in principle feasible by the present photo-responsive capsule. However, the main focus of this study lies on the *photo-controlled release* of encapsulated guests. We believe that the current manuscript contains sufficient examples (7 guests) to convincingly demonstrate the broad applicability of this system in terms of guest size and shape. On the other hand, we are also interested in the maximum size of guest molecules encapsulated by our capsules in water. We would like to reveal the issue in our next project.

For the comments of Reviewer #3:

... There is no doubt that the authors employed a very “smart” building block to realize such interesting stimuli-responsive capsules, which will receive much attention from the broad readership of both chemistry and materials community. The manuscript is well-organized and the conclusion is very solid. So I strongly recommend it to be published in *Nature Communications* after minor revisions.

We sincerely appreciate having such a positive evaluation of this work from Reviewer #3.

1) The authors claimed that they obtained the structure **2a** in water, which contained spherical assemblies $(o-1a)_n$ ($n = \sim 5$). I am wondering if such pentamer is the main product in water. Is it possible to obtain other capsules such as hexamer or the higher-order structures?

Nanocapsule **2a** is a structurally non-uniform product with a narrow size distribution in water. On the basis of the DLS and AFM analyses, combined with molecular modelling, the pentamer is one of the most likely assemblies in water at a concentration of 1.0 mM (based on **o-1a**). For the structural comparison, we added the optimized structures of the tetramer, pentamer, and hexamer as well as their average diameters to Supplementary Fig. 12.

2. How about the stability of the resultant capsule **2a**? For example, will it disassemble if the temperature or the polarity of solvent is changed?

The stability of an aqueous solution of **2a** (1.0 mM based on **o-1a**) against methanol and higher temperature was investigated and the results were added to the revised text (page 4-5) and SI as follows: “The stability of nanocapsule **2a** against methanol and temperature was investigated using ^1H NMR analysis. Formation of nanocapsule **2a** is mainly derived from the hydrophobic effect and π -stacking interactions so that the addition of polar organic solvents results in dissociation of the capsular structure. A ^1H NMR titration study of a D_2O solution of **2a** (1.0 mM based on **o-1a**) revealed that around 40% CD_3OD by volume are required for complete disassembly into monomers **o-1a** (see Supplementary Fig. 13). The complete disassembly of **2a** was also observed upon heating the solution to around 100 °C (see Supplementary Fig. 14). In both cases, the gradual dissociation is accompanied with a sharpening of the ^1H NMR spectrum and a downfield shift of the lateral anthracene protons, caused by the loss of intermolecular π -stacking interactions.”

3. If the pentamer is the main product, what happens during the disassembly process induced by light? Is it possible that five ortho-dianthrylbenzene units undergo the light-induced structural transformation simultaneously or in a step-by-step procedure? It will be very helpful if the authors can provide more information about such supramolecular transformation.

In order to obtain additional information about the disassembly process, a D_2O solution of **2a** (1.0 mM based on **o-1a**) was irradiated stepwise at 380 nm. The ^1H NMR spectrum obtained after 20 sec indicates the formation of intermediary mixed assemblies $(o-1a)_n \cdot (c-1a)_m$. The broad ^1H NMR signal of the product was slightly downfield-shifted as compared to **2a**, which indicates a reduced degree of the π -stacking interactions, and did not overlap with signals of free **c-1a**. The spectrum indicates an estimated conversion of 50% and does not correspond to a diluted ^1H NMR spectrum of **2a** (0.5 mM based on **o-1a**), which excludes a mere concentration effect. Thus, we added the corresponding experiment to the revised SI and the following sentence to the revised text (page 5): “Monitoring of the

reaction by means of ^1H NMR analysis furthermore suggested a stepwise disassembly process and the formation of intermediary assemblies $(\mathbf{o-1a})_n \bullet (\mathbf{c-1a})_m$ with a reduced degree of π -stacking interactions in water (see Supplementary Fig. 43).”

4. Two recent reviews are suggested to be cited in the revised version. One is about the topic of supramolecular transformation (*Chem. Soc. Rev*, 2016, 45, 2656). Another one is about the stimuli-responsive functional assemblies (*Acc. Chem. Res.*, 2018, 51, 2699). It will be very helpful for readers to learn the related field.

The suggested two reviews are very helpful and were added as references 12 and 13 to the manuscript.

REVIEWERS' COMMENTS:

Reviewer #1 (Remarks to the Author):

I am satisfied with the response of the authors and the revisions made. I therefore recommend acceptance of this manuscript without further changes.

Reviewer #3 (Remarks to the Author):

This is a revised version resubmitted by Yoshizawa and coworkers. I have checked the updated manuscript very carefully. I can tell that the authors have already revised the paper according to the reviewers' suggestion. The quality of the revised manuscript has a large step compared to the original one. So I strongly recommend that the current version should be accepted by Nature Communications.